# Feasibility of Optical Bearing Fabrication Using Radiation Pressure

**DOI:** 10.3390/mi13050733

**Published:** 2022-05-02

**Authors:** Yasuhiko Arai, Eri Yane, Ryosuke Koyama

**Affiliations:** Department of Mechanical Engineering, Faculty of Engineering Science, Kansai University, 3-3-35, Suita 565-0871, Osaka, Japan; k779762@kansai-u.ac.jp (E.Y.); k250654@kansai-u.ac.jp (R.K.)

**Keywords:** micromachine, light radiation pressure, optical bearing, three-dimensional printer, friction reduction, microfabrication

## Abstract

A three-dimensional (3D) printer was used to create a model device to discuss the reduction in friction generated by rotation and investigate the possibility of friction reduction in microelectromechanical systems (MEMSs) using light as a future technology. Experiments on this model showed that friction could be reduced using the light radiation pressure. In addition, the possibility of reducing the effect of the friction generated during rotation was demonstrated by adding a mechanism to the rotating rotor mechanism that reduces friction based on the radiation pressure. The effectiveness and associated problems of 3D printers as a fabrication technology for MEMSs were explored.

## 1. Introduction

Microelectromechanical systems (MEMSs) incorporate microelements on a single substrate using various nano/microfabrication technologies. Low power consumption, high performance, and low cost are achieved using wafer-level batch processing with certain phenomena together with new analysis and measurement technologies based on quantum mechanics [1,2,3]. Electrical phenomena, such as electrostatic forces, and piezoelectric technologies, such as piezoelectric elements, have been used as driving power sources for MEMSs [3]. However, electrical driving requires fabrication techniques, such as those for micro-wiring and electromagnetic noise, that can precisely create microstructures. Furthermore, according to the driving principles of mechanical and electrical elements based on various scientific technologies, achieving the intended functions of MEMSs by simply downsizing the driving principles of the conventional size to a micro area without changing the conventional structure is challenging [3].

A typical case is the approach to friction [3]. Dust suspended in the air has a mass; thus, under Newtonian mechanics, it should immediately fall under the influence of gravity, according to its mass. However, dust does not fall as easily as an apple owing to the effect of friction with the air acting on the surface of the dust and the effect of viscosity. In addition, as a microstructure becomes finer, the Reynolds number decreases, and mechanical elements must be operated in a low-Reynolds fluid, even though they are in the air. The behavior of dust in the air and the condensation of water by the capillary action are examples of peculiar behavior in the microscopic region [4]. As many similar examples exist, MEMSs should be fabricated by considering not only their size difference with mechanical elements according to the conventional methods in our daily lives but also phenomena that differ from the driving principles of structures employed on a daily basis. Therefore, MEMSs must be treated from this perspective [3]. In particular, the friction phenomenon in the microscopic region differs from that in normal-size structures.

In the early 1990s, the fabrication of micromotors based on semiconductor manufacturing was reported as a power source for MEMSs [5,6,7], which would be intensely affected by such friction. Without a sufficient friction reduction for the rotating body, fabricated motors could not effectively function over time owing to adhesive wear [8,9,10] between solid bodies.

In conventional large-scale mechanical elements used in daily applications, the friction of the rotating shafts and other parts is reduced by ball bearings, thrust bearings, etc. However, to use ball bearings or oil/air thrust bearings for mechanical elements with a rotating shaft as MEMSs, a microstructure must be fabricated in accordance with the physical phenomena in the microscopic region [11]. Although studies were conducted on the reduction of friction [12,13,14,15] using bearing balls at sizes of several hundred micrometers, only bearings with balls smaller than the MEMS body can realistically be used as microstructures in MEMSs.

Furthermore, for MEMSs to be used under vacuum conditions in the future (e.g., in outer space), employing bearings that use oil and air based on the conventional driving principle will be challenging. In addition, a mechanism using repulsive forces owing to electrostatic forces is typically considered in MEMSs [15]. However, the use of electrical phenomena may result in the fine design and fabrication of wiring and influence the electromagnetic noise.

In this study, we investigated the possibility of light-based friction reduction as a futuristic technology in MEMS. Indeed, it differs from conventional technologies in that it does not use oil or air as a lubricant.

Light has a force referred to as optical radiation pressure, as reported by Ashkin [16,17,18,19,20,21,22]. In this study, a model to examine the friction reduction by light was fabricated using a three-dimensional (3D) printer, and the possibility of friction reduction in MEMSs using light was examined experimentally.

In addition, some problems are reported when a 3D printer is used to fabricate the models to examine the friction reduction by light.

## 2. Principle

### 2.1. Radiant Pressure

The concept of the radiation pressure of light was extensively investigated by Ashkin, considered in the 17th century by J. Kepler in his discussion of the behavior of comet tails approaching the sun [16], and J. C. Maxwell in his electromagnetic field theory [23]. Moreover, its existence was experimentally demonstrated more than a century ago by P. N. Lebedev in Russia, E. F. Nichols, and C. F. Hull in the United States [16,24]. Under the concept of radiation pressure, a force is applied to the mirror when light is reflected off the mirror.

In this case, according to a previous study [25], if the energy of light is assumed to be E and the speed of light is C, the momentum is expressed as |p| = E/C, while the force is written as F = dp/dt. If the power of light is P and the angle of incidence is *θ*, the magnitude of the generated force can be expressed as
F = (2P/C)*r* × cos(θ)(1)
where *r* = *R* + (1 − *R*)α/2, *R* is the reflectivity of the mirror, and α is the fraction of the non-reflected light absorbed by the mirror.

Furthermore, Ashkin used Stokes’ law and reported that the force magnitude owing to the radiation pressure can be explained as a physical phenomenon based on the relationship between the velocity of a sphere in a viscous fluid and the force owing to the radiation pressure [20]. In addition, the specific magnitude of the radiation pressure was measured.

For example, when a platinum-coated silica sphere with a diameter of 10 μm is irradiated at 100 mW in water, as presented in the previous study [26], the silica moves at a speed of 179.2 μm/s. In addition, according to Stokes’ theorem, a force of 15.0 pN is applied. When this radiation pressure is applied to an 8-tooth rotor with a diameter of 100 µm by irradiating light with a total power of 53 mW from two directions, the rotor rotates at a rotational velocity of 8.33 rpm (0.87 rad/s). When light is irradiated onto an object in this manner, force is generated on the surface of the reflected object. This phenomenon is used to develop a new technology to reduce the friction between the contacting surfaces of MEMSs and achieve an optical MEMS [27,28,29].

### 2.2. Optical Bearing for Friction Reduction on the Side Wall of a Rotating Shaft

In this study, we first examined the possibility of reducing friction using the light pressure for structures as large as 100 μm using the structure presented in Figure 1, which models the occurrence of friction.

In the experiments, a 3D printer was used to fabricate a micromechanical element, in which the outer wall of the shaft contacted the inner wall of a rotating ring that rotated around the shaft. The rotating ring generates friction with the base floor, as shown in Figure 1.

As illustrated in Figure 1, a hollow ring (rotor) with a hole of diameter, dh, in a disk of diameter, dr, is inserted around a fixed shaft of diameter, ds.

Light is incident on the circumferential gap (ε1) between the inner wall of the hollow ring and the outer wall of the shaft. The structure is designed to reduce the friction between the two walls owing to the radiation pressure generated by multiple reflections within the gap. As for the friction on the contact surfaces in the vertical direction, the structure was designed to lift the bottom of the rotating ring from the base by generating a radiation pressure. To this end, light is irradiated to the vertical gap (ε2) between the bottom of the ring and the floor surface, and multiple reflections are implemented to reduce the friction in the vertical direction of the hollow ring.

The cross-section of the fabricated structure, as shown in Figure 2, is used to illustrate the propagation paths of light.

To reduce friction in the circumferential direction, light incident downward from the top of the shaft by bending by mirror-1 (3) is split to the left and right by triangular mirror-2 (5), as shown in Figure 2a. Each light irradiates to the inner wall of the ring through aperture-1 (4) by passing through the light path fabricated in the shaft. These lights are then multiplied when reflected by the circumferential gap (1) between the inner wall of the ring and the outer wall of the axis of rotation and then spread over the circumference of the ring.

Moreover, from the floor surface of the base, as shown in Figure 2b, the light incident from the side of the base passing through the light path in the base is bent upward by mirror-3 (7) and irradiated through aperture-2 (6) on the floor surface to the vertical gap (ε2) between the bottom of the ring and the floor surface for multiple reflections. Light from the base was used to reduce the friction on the floor of the ring.

In this case, light from the floor surface enters the ring from two entrances (9), as shown in Figure 2d. Ring (2) is supported upward by the radiation pressure from three mirrors-3 (7) placed at three points on the floor surface. We investigated the possibility of friction reduction by the radiation pressure using a model with such a structure in the order of approximately 100 µm. The evaluation method for the friction reduction state is as follows:

A hand (4) is constructed in advance at point P0 on the circumference of the rotating ring (2), as shown in Figure 1, where a micro-torque wrench (3) is connected to point P0, and point P1 at the other end of the micro-torque wrench is moved tangentially by a piezoelectric element. The friction reduction owing to the radiant pressure was evaluated by measuring the deflection of the torque wrench at the moment when the ring began to rotate and comparing torques required for rotation with and without the radiation pressure. Therefore, the possibility of manufacturing bearings using the radiation pressure was investigated.

The torque based on the static frictional force in the dry state, which occurs between the two contacting surfaces, was observed using this measurement method.

### 2.3. Optical Bearing Design

#### 2.3.1. Light-Guiding Structure to the Inner Wall of the Rotating Ring

As shown in Figure 2a, light is guided downward parallel to the axis of rotation from mirror-1 (3) and emitted to the inner wall of the rotating ring (2) using mirror-2 (5) installed inside the shaft. Finally, as an angle was set between the emitted light and the inner wall of the rotating ring (2), the structure was designed to spread the radiated light around the circumference with multiple reflections.

With such multiple reflections, for example, when the center and axis of rotation of the rotating ring (2) are misaligned, the center of rotation is misaligned when the distance between the inner wall of the rotating ring (2) and the axis of rotation is larger on the outwardly displaced side and smaller on the opposite side. Consequently, the number of multiple reflections increases when the misalignment is narrower and decreases in other directions. Thus, the radiation pressure force is larger at narrower gaps than that at wider gaps, which is considered to cause the self-alignment of the bearing. It is expected that a bearing with a self-alignment capability can be realized.

As shown in Figure 2c, the cross-sectional structure of the model was designed and fabricated such that the rotating ring (2) was separated from the shaft (1) by support (8) at the time of fabrication. The inner wall of the rotating ring (2) and outer wall of the shaft (1) are not in contact at the stage of fabrication using the 3D printer. When the rotating ring (2) is used, by removing the support (8), the ring (2) is inserted into the shaft (1), and the two surfaces contact each other.

#### 2.3.2. Light-Guiding Structure to the Bottom of the Rotating Ring

As shown in Figure 2b, light is introduced from the side to the base of the friction-reduction confirmation mechanism and irradiated to the bottom of the rotating ring (2) by mirror-3 (7) on the base, where it is multiplied, reflected, and spread.

As indicated by the horizontal cross-section of the base in Figure 2d, the base has two light entrance points (9). The light incident from the two points (9) is designed to reach the bottom of the rotating ring (2) through three apertures (6) and (7) at intervals of 120°. The ring was designed such that the balanced radiation pressure from the three locations pushed the rotating ring (2) upward.

Based on the above design strategy, the friction-reduction confirmation mechanism was designed in this study using Autodesk Inventor as a 3D computer-aided design (CAD) software.

### 2.4. Fabrication of a Friction-Reduction Confirmation Mechanism

In this study, a friction-reduction confirmation mechanism was fabricated using a Nanoscribe Photonic Professional GT with a fabrication resolution of 200 nm as the 3D printer and an IP-Dip resist as the structural material.

Because the movable range of the 3D printer was 300 µm, the diameter of the model shaft was set to 80 µm, as shown in Figure 1, whereas the outer and inner diameters of the rotation ring (2) were set to 180 and 90 µm, respectively; thus, the experiment could be performed within the field of view of an optical microscope. The gap in the circumferential direction between the inner wall of the rotation ring (2) and the outer wall of the shaft (1) was set to 5 µm.

However, as the 3D printer could not fabricate a gap of 2 µm or smaller, as shown in Figure 3a, the rotating ring (2) was fabricated with supports (8) in four directions such that it could be separated by supports (8) upon fabrication using the 3D printer. After the walls of the structure were processed to a mirror surface (aluminum film thickness = 100 nm) by aluminum sputtering, the supports (8) were removed, and the rotating ring (2) was assembled by dropping it onto the shaft (1). A gap width of 5 μm was consistently achieved using this fabrication process. The experimental apparatus was fabricated, as shown in Figure 3b, by CASTEM Inc. (Hiroshima, Japan) Scanning electron microscopy (SEM) images confirmed that the 3D printer fabricated the structure, according to the CAD data presented in Figure 3a.

## 3. Friction-Reduction Confirmation Experiment

### 3.1. Fabricated Friction-Reduction Confirmation Mechanism

The experimental apparatus, as shown in Figure 3b, was fabricated using a 3D printer using the 3D CAD data presented in Figure 3a. Figure 3b shows a top view of the friction-reduction confirmation mechanism. There are two light entrance points (5) on the base (10). Mirror-1 (5) is introduced on the top for light entrance and propagation in the circumferential direction of the ring (2). To prevent the rotating ring (2) from dissipating after the support (8) is removed, a ring stopper (9) with a six-directional projection was introduced at the top. The experimental apparatus was mounted on the glass substrate of the 3D printer used in the fabrication process, and the base was fixed with an adhesive. In addition, to generate radiation pressure over the entire area of the device, propagate light, and increase the strength of the structure, the entire structure was sputtered with aluminum (film thickness of approximately 100 nm) to create a mirror finish.

Moreover, the rotating ring (2) supported from four directions, as presented in Figure 3b, was manufactured according to the design. Subsequently, the supports were removed, the rotating ring (2) was dropped into the shaft (1), and the device was assembled to form a structure that allowed rotation of only the rotating ring (2). After manual confirmation under a microscope, the rotating ring (2) was rotated, and the state of reduced friction with and without the radiation pressure was checked.

### 3.2. Experimental Apparatus and Method for Confirmation of the Friction Reduction

In the friction-reduction experiment, friction reduction with and without the light radiation pressure was observed using an observation device, as outlined in Figure 4. The friction-reduction confirmation mechanism was fixed on the table. The light was introduced from the upper and lower light inlets using a fiber (Lensed Tip Fiber Patch Cable; Thorlabs).

Based on the camera images, as shown in Figure 5a, a micro-torque wrench (3) composed of an ultrafine platinum wire with a diameter of 625 nm (The Nilaco Corporation) was attached to the four-way protrusion (hand) (4) of the rotating ring (2) that was removed from the support, as shown in Figure 5b. Furthermore, Figure 5a demonstrates that a piezoelectric actuator, manufactured by PI, was used to rotate the hand at a constant speed in the tangential direction. The speed was observed at 27.5 μm/s because of the relationship between the frame rate of the used camera and the pixel size of the camera. Rotational torque was applied to the rotating ring (2) using a micro-torque wrench (3). In this case, the change in deflection of the micro-torque wrench at the start of the rotation of the rotating ring with and without incident light was recorded as a movie. Subsequently, the deflection of the torque wrench was determined by measuring the number of pixels related to the deflection of the micro-torque wrench on the image. The torque applied to the device was determined from the deflection obtained. The friction-reduction status was confirmed by changes in torque with and without the optical radiation pressure.

In the experiment, the micro-torque wrenches were broken several times; thus, we used microcantilevers with a length in the range of 185.6–237.8 μm to determine the contact point of the micro-torque wrench with the device. The length was measured by counting the pixels of the camera (Table 1) and detailed observation of the contact point between the tip of the micro-torque wrench and the projection of the rotating ring. In the experiment, a green laser (wavelength = 532 nm, output power = 5 W) with a wavelength that is relatively easy to obtain and capture with a camera was branched using a bifurcated optical coupler and irradiated from the entrance of light (5) at three upper and lower locations, as shown in Figure 3b. The fiber output at each entrance was measured using a power meter. The experiment was conducted in the air. The laser power supplied by the fiber was adjusted such that the optical output was 40 mW from the top entrance and 15 mW from each aperture of the three lower locations on the base.

Initially, a laser power of approximately 5 mW was irradiated from each incident light. However, no change in torque related to the presence or absence of light was observed when using the micro-torque wrench. Although the same torque remained approximately independent of the presence or absence of light up to approximately 20 mW, the starting rotational torque changed with the presence or absence of light when the light was irradiated in the circumferential direction beyond 30 mW. Therefore, we decided to conduct this experiment under an incident power of 40 mW. However, as the experiment was carried out in the air, it was not possible to carry out the experiment with light energy higher than 40 mW because the excessive laser input caused damage even to the aluminum-sputtered experimental apparatus.

In the future, we will need to modify the intensity of the device to provide higher power, collect data on the light power and friction force improvement, and obtain results of the 40 mW case to determine the optimal light input power. However, in this study, under irradiation values of 40 mW in the circumferential direction and 15 mW (each) from the lower direction, a reduction in frictional torque could be confirmed.

### 3.3. Experimental Results

The experimental results are listed in Table 1. The micro-torque wrenches were damaged several times and experiments were carefully numbered to avoid any data confusion. When the wrench was damaged, new micro-torque wrenches were sequentially fabricated and evaluated in five experiments, with and without light incidence. Although the amount of deflection generated varied with the length of the micro-torque wrench used in each experiment, the standard deviation of the rotational torque generated with and without light was measured to be 0.28 and 0.26 pNm, respectively, confirming similar variability. The test of variability using the F distribution confirmed that no differences existed in the variability of results between the two sets of data at a confidence level of 95%.

According to the comparison of the cases without and with light incidence, the ring began to rotate with an average torque of 2.22 pNm by the micro-torque wrench when light incidence did not exist. This torque was considered to be generated by the static frictional force.

Moreover, in the case of light irradiation, the ring began to rotate when a rotary torque of 1.21 pNm was applied using a micro-torque wrench. The rotational torque relative to friction was reduced by approximately 45% with light irradiation. A t-distribution test of the difference in mean values with and without light confirmed at a confidence level of 95% that the rotational torque was lower with light incidence.

We believe that a more precise observation will be necessary in the future and the surface roughness and characteristics under vacuum should be analyzed in consideration of future applications.

We experimentally confirmed the possibility of improving the operating efficiency of MEMSs by reducing friction using light radiation pressure.

## 4. Application of the Friction-Reduction Phenomenon Using the Radiation Pressure to a Rotating Rotor

We considered the possibility of friction reduction by light obtained in this study for application in the rotor rotation experiment using the radiation pressure, which is a future technology to supply the power source of MEMSs, as presented in a previous study [26]. Because the friction-reduction experiment was conducted on a rotating rotor, the reduction of dynamic friction in addition to the static friction will be examined.

In a previous study, an eight-bladed rotor (thickness = 10 μm) with a radius of 50 μm, as shown in Figure 6a, rotated with an angular velocity of 0.87 rad/s under light irradiation with a total power of 53 mW from two directions [26].

In this study, a system was fabricated in which light incident from the tangential direction of the shaft, as shown in Figure 6c, was multiply reflected at the gap (3 μm) where the inner wall of the cylinder supports the shaft and outer wall of the shaft contact, as shown schematically by the red arrow, to confirm the friction reduction.

Using this experimental setup, the possibility of reducing the friction of a rotating rotor was investigated by comparing changes in the rotational angular velocity with and without light incidence in the tangential direction. As shown in Figure 7a, the fabricated structure was an eight-blade rotor (rotor thickness = 10 μm, diameter = 70 μm) (2) that was irradiated with light from three directions to rotate the rotor. In this experimental apparatus, as shown in Figure 7a, a light entrance with a diameter of 15 μm was provided at point (3) (bearing section), where the rotating shaft was supported, and the light was incident on the gap (3 μm) between the inner wall of the cylinder that supports the rotating shaft and the outer wall of the rotating shaft. Figure 7b shows an SEM image of the experimental apparatus fabricated using a 3D printer observed from the top. The structure was designed for light incidence from the direction indicated by a thick white arrow to generate a bearing effect in addition to the light to rotate the rotor.

In particular, the friction-reduction effect was investigated by examining the changes in the rotational angular velocity of the rotor when the rotor was rotated with a light power of 50 mW from three directions and light incident on the bearing section at a power of 3 mW, which was expected to have a bearing effect. The case must be analyzed when a high-power light was incident on the bearing area to determine the light intensity that could be expected to have a bearing effect. However, when the light was irradiated through a hole with a diameter of 15 μm in the gap between the shaft and bearing inner wall, the application of a power of 5 mW or higher damaged the bearing components. Thus, this study was limited to experiments that used a maximum power of 3 mW.

However, even with this simple structure, as shown in Figure 7c, when the light was incident, a ring of light was observed around the shaft of the rotor at a position opposite to that of the incident light. This phenomenon is thought to be caused by the multiple reflections of the light incident through a hole in the gap with a diameter of 15 μm. A detailed study of the realization of multiple reflections in the gap will be conducted in the future to investigate more effective friction-reduction conditions.

Under the above conditions, the difference in angular velocity of rotation with and without light was analyzed by movies (frame rate at 1/30 s) of the rotor rotating under a microscope, as shown in Figure 8.

This experiment was conducted under similar conditions in a previous study [26]. To avoid a temperature increase in the device owing to light, the device was immersed in ethyl alcohol and cooled using a Peltier element such that the temperature exactly below the axis of rotation was always maintained below 0 °C during the experiment.

In the case without light incidence, which was expected to have a bearing effect, as shown in Figure 8a, the rotor (3) of the landmark rotated by approximately 25° in 0.40 s, as shown in Figure 8b. The rotor rotates at an average angular velocity of 1.15 rad/s.

Furthermore, a different light beam (the blue arrow) from the driving light was irradiated at a power of 3 mW from the newly installed fiber-2 with a lens to the gap between the shaft and ring. As shown in Figure 8c, experiments were conducted to evaluate the bearing effect when the rotor was rotated with the light beam irradiated in three directions. As shown in Figure 8d, the landmark rotor (3) rotates by approximately 35° in 0.20 s. The rotor rotates at an average angular velocity of 3.12 rad/s. The introduction of light, which is expected to have a bearing effect, roughly tripled the angular velocity of the rotation. However, the power of the light used for reducing the friction was limited to 3 mW.

We could confirm the possibility of friction reduction, even with a simple structure and low input power. This result is significant for the future development of optical bearings. Therefore, we believe that it is possible to develop effective bearings that use light to reduce friction in MEMSs, which used to be difficult.

However, a problem was identified when a 3D printer was used for fabricating the structures. The 3D-printed structures were fabricated using a 3D printer and then processed using aluminum sputtering to improve the mechanical strength of each element. However, during the experimental process, the structure was repeatedly damaged by the laser beam and deformed owing to the low strength of the components, and the friction reduction could not be studied when the light input was more than 3 mW during light irradiation. All problems were caused owing to the fragility of structures fabricated by the 3D printer. In the future, therefore, it is necessary to improve the strength of fabricated structures, investigate means of reinforcement using sputtering technology, and develop appropriate design methods. If the above problems are solved, we believe that 3D printers, which enable precision processing, will become an indispensable fundamental technology for MEMS.

In addition to the development of MEMS power supplies using micro rotors, we plan to conduct further studies on the MEMS processing technology, particularly for the widespread use of 3D printers, which are effective in fabricating microstructures.

## 5. Conclusions

The results of this study can be summarized as follows:(1)To investigate the possibility of friction reduction in MEMSs using light as a future technology, a model was fabricated using a 3D printer to study the friction generated by rotation on a 100 μm square surface. It was confirmed that friction could be reduced using radiation pressure.(2)We fabricated an experimental apparatus that adds a device to promote the friction reduction based on the radiation pressure to a rotating rotor mechanism using light, which is currently being developed as a power source for MEMSs [26]. We used this apparatus to confirm that the effect of friction could be reduced by the radiation pressure, even in the case of dynamic friction generated during rotation.(3)The effectiveness of 3D printers as a fabrication technology for MEMSs was demonstrated. However, to achieve a stable application of structures fabricated by a 3D printer using resist materials, it is necessary to develop a reinforcement method that simulates the skeletal structure of crustaceans in the current aluminum sputtering technology (thickness = 100 nm), reinforcement method using metals other than aluminum, and reinforcement method to improve the strength of the structure.

## Figures and Tables

**Figure 1 micromachines-13-00733-f001:**
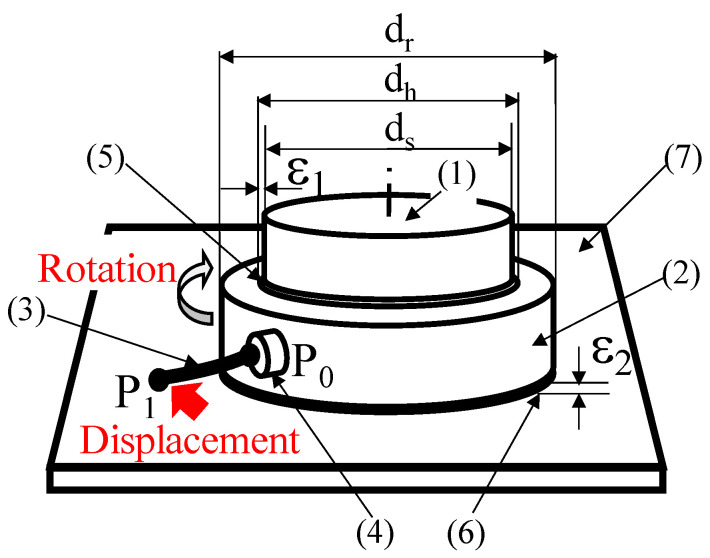
Concept of friction-reduction confirmation mechanism. (1) Shaft (d_s_), (2) Rotating ring, (3) Torque wrench, (4) Hand, (5) Circumferential gap, (6) Vertical gap, (7) Base.

**Figure 2 micromachines-13-00733-f002:**
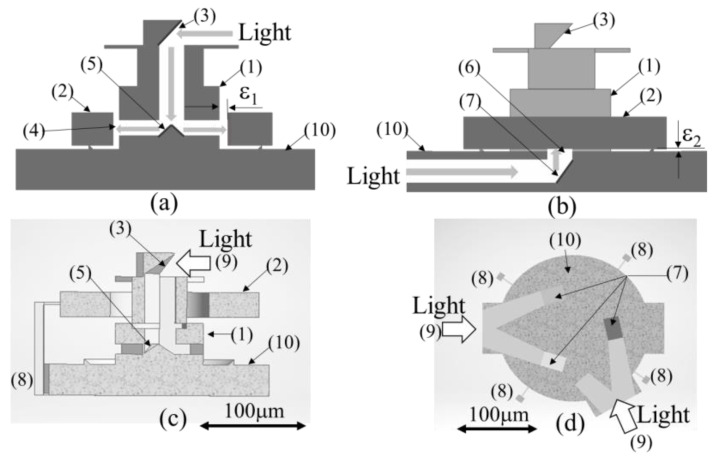
Three-dimensional structure of friction-reduction confirmation mechanism. (1) Shaft, (2) Rotating ring, (3) Mirror-1, (4) Aperture-1, (5) Mirror-2, (6) Aperture-2, (7) Mirror-3, (8) Support, (9) Entrance of light, (10) Base, (**a**) Light propagation in circumferential direction, (**b**) Light propagation in vertical direction, (**c**) Vertical cross-section of model, (**d**) Horizontal cross-section of model.

**Figure 3 micromachines-13-00733-f003:**
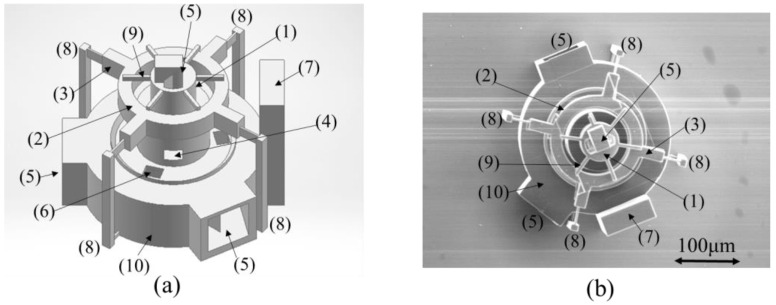
Friction-reduction confirmation experimental apparatus. (1) Shaft, (2) Rotating ring, (3) Hand, (4) Aperture-1, (5) Entrance of light, (6) Aperture-2, (7) Fiber stand, (8) Support, (9) Ring stopper, (10) Base, (**a**) Three-dimensional CAD model, (**b**) Scanning electron microscope (SEM) image of experimental apparatus.

**Figure 4 micromachines-13-00733-f004:**
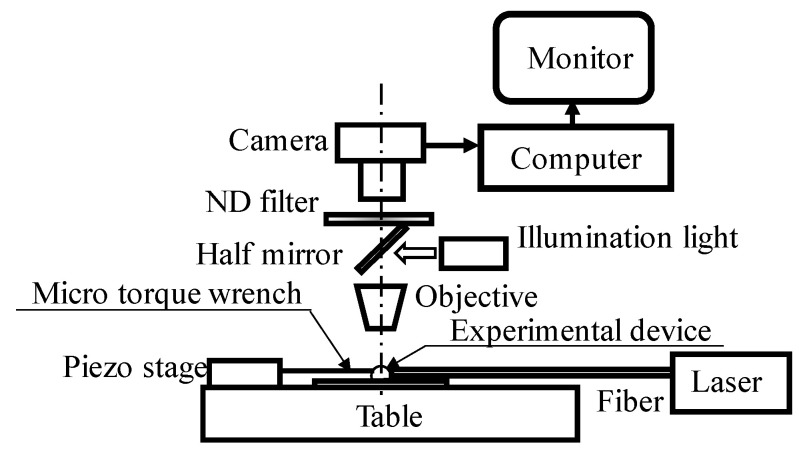
Overview of the experimental measurement system.

**Figure 5 micromachines-13-00733-f005:**
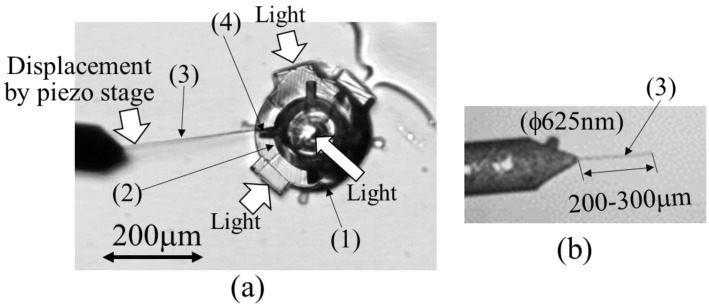
Torque-sensing system: (1) Experimental apparatus, (2) Rotating ring, (3) Micro-torque wrench, (4) Hand, (**a**) Torque-sensing using micro-torque wrench, (**b**) Micro-torque wrench.

**Figure 6 micromachines-13-00733-f006:**
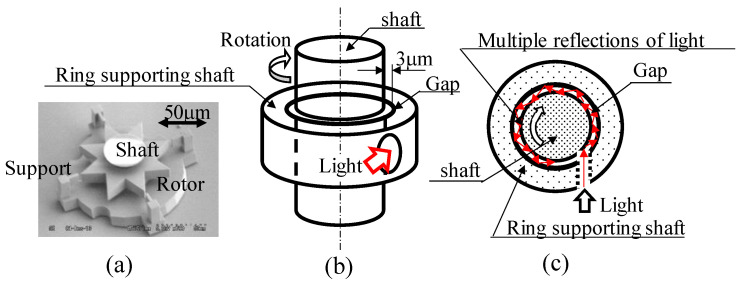
Application of light-based friction-reduction mechanism to micro rotor: (**a**) Micro rotor fabricated by 3D printer, (**b**) Concept of light introduction method for friction reduction, (**c**) Multiple reflections of light in gap between shaft and rotating ring (top view).

**Figure 7 micromachines-13-00733-f007:**
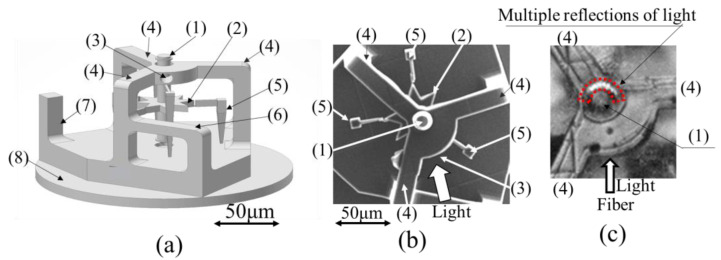
Rotor shaft friction-reduction model: (1) Shaft, (2) Rotor, (3) Entrance of light, (4) Beam, (5) Support, (6) Fiber stand, (7) Fiber partition, (8) Base, (**a**) Three-dimensional CAD model, (**b**) SEM image of experimental apparatus (top view), (**c**) Confirmation of multiple reflections of light on the shaft gap.

**Figure 8 micromachines-13-00733-f008:**
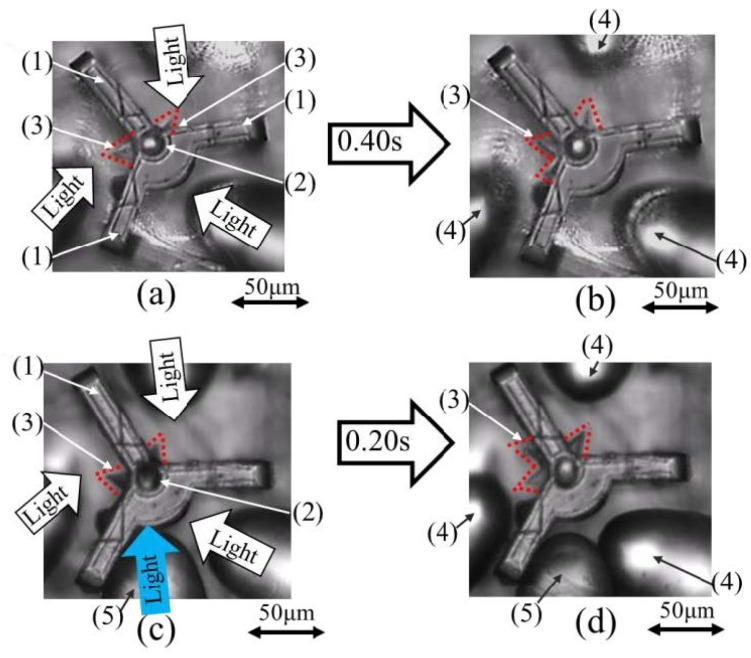
Experiment result of friction reduction: (1) Beam, (2) Shaft, (3) Blade of rotor, (4) Lensed fiber-1 for rotation, (5) Lensed fiber-2 for friction reduction, (**a**)→(**b**) Rotation without light (ω = 1.15 (rad/s)), (**c**)→(**d**) Rotation with light (ω = 3.12 (rad/s)).

**Table 1 micromachines-13-00733-t001:** Experimental results.

		Measurement Results					Average
Without Light	Length of torque wrench (μm)	185.6	231.4	232.3	236.6	117.4	-
	Deflection (μm)	50.0	82.4	81.3	79.9	66.3	72.0
	Torque (pNm)	2.66	2.27	2.20	2.05	1.92	2.22
With Light	Length of torque wrench (μm)	236.5	237.8	235	233.5	228.23	-
	Deflection (μm)	35.0	51.1	47.8	38.9	54.9	45.5
	Torque (pNm)	0.899	1.29	1.25	1.04	1.57	1.21

## Data Availability

Not applicable.

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
