# Peer review of "Feasibility of Optical Bearing Fabrication Using Radiation Pressure"

_micromachines, 2022, doi:10.3390/mi13050733_

Round 1
Reviewer 1 Report
The authors studied the possibility of friction reduction in MEMSs using light. They created a model using a 3D printer to study the friction generated by rotation on a 100-μm square surface. Writing requires improvement through the paper as it is difficult to follow what the authors tries to say.
For example in the abstract: “A three-dimensional (3D) printer was used to create a model device to discuss the friction generated by rotation to investigate the possibility of friction reduction in microelectromechanical systems (MEMSs) using light as a future technology” how can the authors
To discuss not sure why
Rotation of what that creates friction.
Light as a future technology: in what sense?
Another example in the introduction, “ Microelectromechanical systems (MEMSs) incorporate tiny elements on a single substrate using various nano/microfabrication technologies to achieve a low power consumption, high performance, and low cost by wafer-level batch processing with phenomena and new approaches of analysis and measurement technology based on quantum me-chanics.[1–3]”
Tiny: not academically appropriate
The techniques used in mems not all based on quantum mechanics so not scientifically correct.
These low quality writing is all over the paper ( the principle, friction reduction, etc. ).
Text in figures can be reduced.
Results and the idea are promising so I suggest to proof read the paper and resubmit.
Reviewer 2 Report
Submitted manuscript is interesting, timely addressed and it is in the score of the journal. The problem of the friction in MEMs devises is important and widely discussed in the literature. For example, the monolayer covering with this respect can be important as well as the capilarily forces (DelRio, de Boer et al., Model and measurement Applied Physics Letters (2007)). However, the referencing of this interesting study is quite poor and some aspects are referenced without care. For example, the hypothesis of the existence of light pressure was first put forward by I. Kepler in the 17th century due to the need to explain the behavior of comet tails during their travelling near the Sun. In 1873, Maxwell gave a theory of light pressure within the framework of his classical electrodynamics. Experimentally, light pressure was first studied by P. N. Lebedev in 1899. Why should we refer to E. E. Nichols; G. F. Hull, The pressure due to radiation, Phys. Rev. 17, 25 (1903) only and do not take into account previous contributions?
Despite the fact that manuscript has a lot of experimental data they are very difficult to read and understand because of technical deficiencies of the manuscript. The first one is – very short figure captions, which are not informative. For instance, Fig. 1 has 7 parameters, which are not defined before the figure appearance. Figures use different font size for writings (without special reason for it). White writings are almost invisible. Figures must be corrected, they can be bigger and without details overlapping, some writings can be given as 1, 2, 3, … numbers and their description can be given in the figure caption.
What is the meaning of numbers 1,2,3,4,5 for Table 1?
Figure caption for Figure 6 is not clear. “Multiple reflections of light in gap 321 between shaft and rotating ring (Top view)” – is it experiment or simulation?
It is not clear for general reader how the multiple reflections on shaft confirmed by the Figure 7 (c) image?
The last conclusion “The effectiveness of 3D printers as a fabrication technology for MEMSs was demonstrated. However, it was found that reinforcement by aluminum sputtering (100 nm thick) was not sufficient to ensure stable use of structures fabricated by the 3D printer using a resist as the material, and that it would be challenging to widely use the current 3D printer's technology alone in MEMSs in the future. It was found that fabrication technology using a 3D printer needs to be improved to fabricate the stable products” needs additional comments about possible ways of the improvements (may be Ti can be used instead of Al or whatever).
Round 2
Reviewer 2 Report
Authors made special efforts and significantly improved the manuscript. However, they did not take into account the historical part of the effect answering the question as follows: "We would like to handle the bibliography in that manner." This is their right to act in this way. At the same time I cannot agree with such a solution and request additional "minor" revision. In any case, academic Editor can make decision on the basis of all reviewers opinions.